# ENAHPool: The Edge-Node Attention-based Hierarchical Pooling for Graph Neural Networks

**Zhehan Zhao**[1] **Lu Bai**[1] **Lixin Cui**[2] **Ming Li**[3,4] **Ziyu Lyu**[5] **Lixiang Xu**[6] **Yue Wang**[2] **Edwin R. Hancock**[7]

## Abstract

Graph Neural Networks (GNNs) have emerged as powerful tools for graph learning, and one key challenge arising in GNNs is the development of effective pooling operations for learning meaningful graph representations. In this paper, we propose a novel Edge-Node Attention-based Hierarchical Pooling (ENAHPool) operation for GNNs. Unlike existing cluster-based pooling methods that suffer from ambiguous node assignments and uniform edge-node information aggregation, ENAHPool assigns each node exclusively to a cluster and employs attention mechanisms to perform weighted aggregation of both node features within clusters and edge connectivity strengths between clusters, resulting in more informative hierarchical representations. To further enhance the model performance, we introduce a Multi-Distance Message Passing Neural Network (MD-MPNN) that utilizes edge connectivity strength information to enable direct and selective message propagation across multiple distances, effectively mitigating the over-squashing problem in classical MPNNs. Experimental results demonstrate the effectiveness of the proposed method.

## 1. Introduction

Convolutional Neural Networks (CNNs) are powerful tools for processing grid-structured data, such as text, images, and time series. However, real-world data in many domains often involve complex relationships that cannot be easily captured by regular grid formats, such as those found in social networks and molecular structures. To effectively handle such irregular data, which are typically modeled as graphs, Graph Neural Networks (GNNs) have been introduced. In recent years, GNNs have been widely adopted in various fields, including knowledge graphs (Schlichtkrull et al., 2018), drug discovery (Sun et al., 2020), and recommendation systems (Wu et al., 2023).

One challenge arising in GNNs is extracting meaningful representations for graph-level tasks, such as graph classification (Errica et al., 2020) and graph regression (Bianchi et al., 2020). To address this challenge, various graph pooling operations have been proposed, which can generally be categorized into two categories. The first category consists of global pooling methods, which aggregate all node embeddings collectively. However, these methods fail to capture the hierarchical structures inherent in graphs.

To address this issue, hierarchical pooling methods have been developed to preserve the hierarchical structure of graphs by progressively reducing their size. This reduction is typically achieved using two strategies (Ju et al., 2024). The first is the Top-K strategy, which ranks nodes based on a scoring function and retains only the Top-K nodes and their associated edges as the coarsened graph. In contrast, the cluster-based strategy groups nodes into clusters and generates the coarsened graph by aggregating both node features within clusters and edge connectivity strengths between clusters. Since the Top-K strategy suffers from information loss due to the direct removal of nodes and edges, the cluster-based strategy is generally considered more effective.

Unfortunately, many cluster-based methods rely on soft node assignment (Ying et al., 2018), where node features are probabilistically assigned to different clusters and then summed to form cluster representations. This approach can lead to influential nodes being split across multiple clusters, while less influential nodes are assigned entirely to a single cluster, potentially causing the latter to dominate the cluster representations. Although some alternative methods, such as SEP-G (Wu et al., 2022), can assign nodes to single clusters, they still employ simple summation for aggregation,

---

[1]School of Artificial Intelligence, Beijing Normal University, Beijing, China. [2]School of Information, Central University of Finance and Economics, Beijing, China. [3]Zhejiang Institute of Optoelectronics, Jinhua, China. [4]Zhejiang Key Laboratory of Intelligent Education Technology and Application, Zhejiang Normal University, Jinhua, China. [5]School of Cyber Science and Technology, Sun Yat-Sen University, Shenzhen, China. [6]School of Artificial Intelligence, Hefei Institute of Technology, Hefei, China. [7]Department of Computer Science, University of York, York, United Kingdom. Correspondence to: Lu Bai <bailu@bnu.edu.cn>.

*Proceedings of the $42^{nd}$ International Conference on Machine Learning*, Vancouver, Canada. PMLR 267, 2025. Copyright 2025 by the author(s).

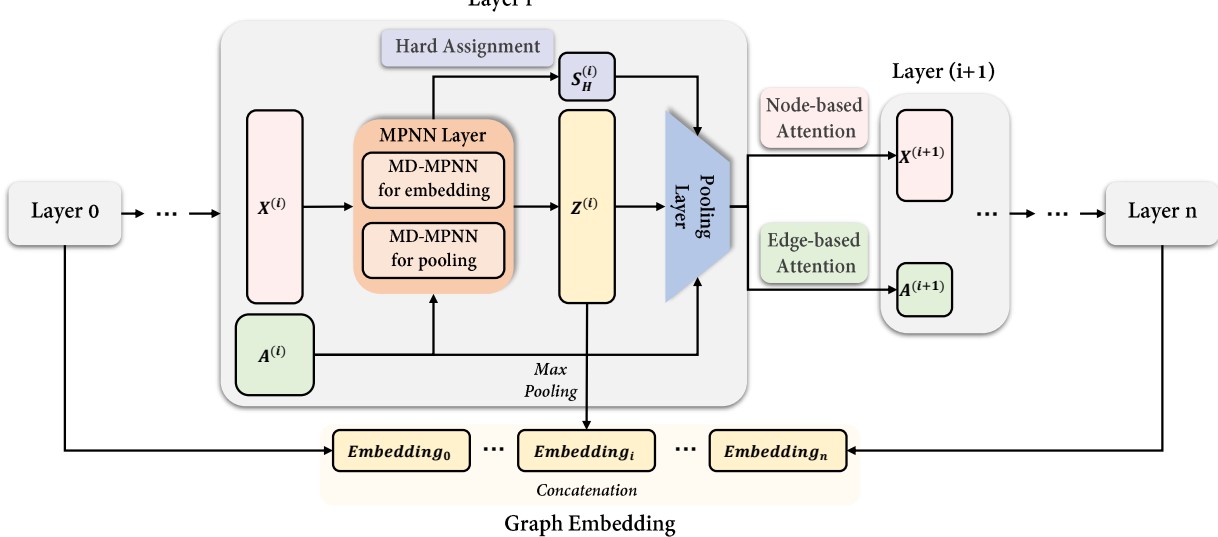

*Figure 1.* The framework of the proposed ENAHPool operation associated with the MD-MPNN architecture.

overlooking the importance of individual nodes. To address this limitation, several studies (Liu et al., 2022; Ye et al., 2023) have introduced attention mechanisms to aggregate node features within clusters. However, these approaches are computationally expensive, due to the fact that they lack the ability to process graphs in a parallel way.

On the other hand, when performing convolution on the coarsened graphs generated by cluster-based methods, edge connectivity strengths can naturally serve as attention weights, guiding selective information aggregation. However, existing cluster-based methods typically divide each edge into different cluster pairs based on the assignment probabilities of its connected nodes, and aggregate the connectivity strengths between identical cluster pairs via simple summation. This overlooks the importance of individual edges, resulting in aggregated edge connectivity strengths that fail to accurately reflect the influence between clusters. For example, for a social network, the edge connectivity strengths between the leadership teams of Companies A and B may be equal to that between the employees of Companies A and C, but the influence of B and C on A could be significantly different. Therefore, it is essential for cluster-based methods to accommodate the importance of edges when aggregating their connectivity strengths.

Furthermore, most graph pooling operations are integrated with Message-Passing Neural Networks (MPNNs) (Gilmer et al., 2017), which often suffer from the notorious over-squashing problem (Topping et al., 2022; Giovanni et al., 2023) and influence the effectiveness of pooling methods (e.g., Figure 2). This drawback arises because MPNNs primarily rely on the iterative aggregation of information from neighboring nodes to capture signals from distant parts

of the graph. As a result, the receptive field of each node grows exponentially, incorporating a large amount of redundant information from its neighbors. Eventually, the information from such an exponentially growing receptive field is squashed into a fixed-size representation, severely limiting the ability of MPNNs to capture long-range information (Alon & Yahav, 2021; Dwivedi et al., 2022).

To overcome this limitation, some graph rewiring strategies have been developed. One category comprises local rewiring methods, which allows direct message passing between nodes at different distances (Abboud et al., 2022; Gutteridge et al., 2023; Ding et al., 2024). However, these methods treat all nodes at the same distance equally, ignoring the actual edge connectivity strength information among them. Another strategy is based on global rewiring, such as graph Transformers (Ying et al., 2021; Zhang et al., 2023), which construct fully connected graphs and apply attention mechanisms for information aggregation. However, these approaches tend to disregard the original graph topology.

The aim of this work is to address the aforementioned shortcomings by introduce a novel Edge-Node Attention-based Hierarchical Pooling (ENAHPool) operation, that is proposed based on the Multi-Distance MPNN (MD-MPNN) architecture. The framework of the proposed method is illustrated in Figure 1, and the main contributions are threefold.

**First**, we introduce a novel cluster-based hierarchical pooling operation, namely ENAHPool, that assigns each node exclusively to a cluster and incorporates attention mechanisms to identify the importance of nodes within clusters and edges between clusters. This design addresses the limitation of uniform edge-node information aggregation arising in existing cluster-based methods and results in more

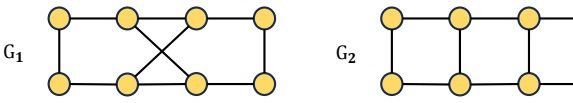

*Figure 2.* An example of the expressive limitations of conventional MPNNs, where $G_1$ and $G_2$ cannot be distinguished.

informative hierarchical representations. **Second**, we propose a Multi-Distance MPNN architecture that works in conjunction with the proposed ENAHPool. This architecture enables direct message propagation between nodes at different shortest distances, effectively mitigating the over-squashing problem in conventional MPNNs. Moreover, it leverages edge connectivity strength information to selectively aggregate information from equidistant nodes, further enhancing the representational capacity of the architecture. **Third**, we evaluate the classification performance of the proposed ENAHPool operation associated with the MD-MPNN architecture, the experiments demonstrate its effectiveness.

## 2. Related works

### 2.1. Graph Pooling

Graph pooling operations typically classified into two main categories. The first category consists of global pooling operations, which aggregate information from all node embeddings in a single step, commonly known as read-out functions. Such operations can be implemented using permutation-invariant functions (Duvenaud et al., 2015; Xu et al., 2019) or neural networks (Vinyals et al., 2016; Zhang et al., 2018; Baek et al., 2021). However, these methods overlook the inherent hierarchical structure characteristics of graphs (Knyazev et al., 2019; Bianchi & Lachi, 2023).

To overcome this problem, hierarchical pooling operations have been developed. These operations can capture the hierarchical structures of the graph by progressively coarsening it, which is achieved through the TopK-based strategy and the cluster-based strategy. Nevertheless, these hierarchical pooling methods still rely on readout functions to extract graph representations from the coarsened graphs.

The TopK-based strategy involves learning the importance values of nodes and selecting the Top-K nodes, along with their corresponding edges, to form the coarsened graph (Gao & Ji, 2019; Lee et al., 2019). However, such methods often discard many nodes and edges, resulting in the loss of valuable information within the graph (Liu et al., 2023).

The cluster-based strategy progressively compresses the graph by clustering nodes while preserving the connections between clusters. Existing cluster-based methods primarily focus on optimizing the clustering process. For example,

DiffPool (Ying et al., 2018) employs GNNs to generate assignment matrices, while StructPool (Yuan & Ji, 2020) captures higher-order structural relationships to guide node assignments. MinCutPool (Bianchi et al., 2020) applies spectral clustering for node grouping, and SEP-G (Wu et al., 2022) minimizes structural entropy to construct a hierarchical coding tree from nodes. DMoN (Tsitsulin et al., 2023) leverages the modularity measure to evaluate clustering quality, whereas WGDPool (Xiao et al., 2024) introduces a differentiable k-means clustering mechanism. However, these methods perform uniform aggregation by summing node features within clusters and edge connectivity strengths between clusters, without considering the varying importance of individual nodes and edges. Although some methods, such as ABDPool (Liu et al., 2022) and C2N-ABDP (Ye et al., 2023), incorporate attention mechanisms for node aggregation, they still overlook the importance of edges.

### 2.2. Message-Passing Neural Networks

Message-Passing Neural Networks (MPNNs) (Gilmer et al., 2017) are a widely used framework for graph-related tasks. The message passing process consists of two main steps: (1) Initialization: node embeddings are initialized using their attributes or predefined rules, such as node degrees; and (2) Updating: node embeddings are iteratively updated by aggregating the embeddings of their adjacent nodes. After $k$ iterations of message passing, corresponding to $k$ layers in the network, each node incorporates information from its $k$-hop neighborhood. This process can be mathematically written as (Ju et al., 2024)

$$h_u^{(l)} = \text{UPDATE}(h_u^{(l-1)}, \text{AGGREGATE}(\{h_v^{(l-1)} : v \in \mathcal{N}(u)\})).$$

where $h_u^{(l)}$ is the embedding of node $u$ after being updated by the $l$-th layer, where $h_u^{(0)}$ is the original node feature and $\mathcal{N}(u)$ denotes the set of adjacent nodes of node $u$. This message-passing paradigm has been adopted in many influential works, e.g., GCN (Kipf & Welling, 2017), Graph-SAGE (Hamilton et al., 2017) and GIN (Xu et al., 2019).

However, this iterative message-passing paradigm gives rise to the over-squashing problem, which severely limits the capacity of MPNNs to capture long-range dependencies (Topping et al., 2022). To mitigate this issue, MixHop (Abu-El-Haija et al., 2019a;b) utilizes powers of the normalized adjacency matrix to directly aggregate information within $k$-hop neighborhoods. The shortest-path-aware MPNNs, such as SPN (Abboud et al., 2022) and GRED (Ding et al., 2024), directly aggregate information from nodes at different shortest path lengths, further alleviating the over-squashing issue. In addition, graph Transformer models, such as Graphormer (Ying et al., 2021) and GPS (Rampásek et al., 2022), can inherently capture global context, addressing the challenge of long-range dependency modeling in conventional MPNNs.

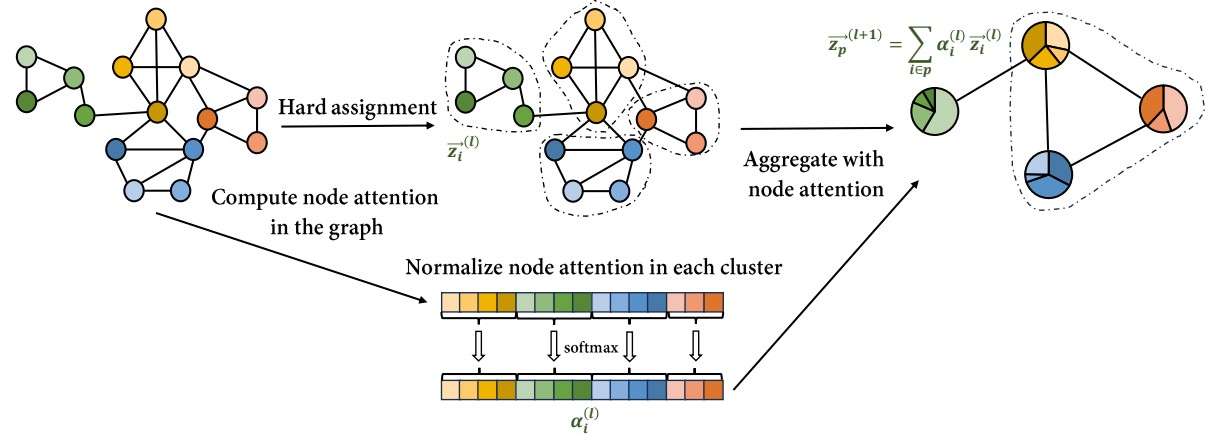

*Figure 3.* The process of hard node assignment and node aggregation using node-based attention mechanism.

However, these approaches generally overlook the edge connectivity strength information in the graph. Notably, the edge connectivity strengths in coarsened graphs generated by cluster-based methods reflect the influence of information propagation between clusters. Therefore, existing MPNNs are not well-suited for cluster-based pooling methods, as they fail to exploit this critical structural information.

## 3. The Proposed Methods

### 3.1. Preliminaries

We denote the input graph as $G^{(0)}(V^{(0)}, E^{(0)})$, where $V$ represents the nodes and $E$ represents the edges. Since hierarchical pooling operations alter the number of nodes at each pooling layer, we define the graph at the $l$-th layer as $G^{(l)}(V^{(l)}, E^{(l)})$. The connectivity strengths between nodes in $G^{(l)}$ are represented by an adjacency matrix $A^{(l)} \in \mathbb{R}^{N_l \times N_l}$, where $N_l = |V^{(l)}|$ is the number of nodes at $l$-th layer. The node feature matrix is denoted as $X^{(l)} \in \mathbb{R}^{N_l \times d_l}$, where $d_l$ is the feature dimension at $l$-th layer.

### 3.2. The ENAHPool Operation

An overview of the proposed ENAHPool operation is illustrated in Figure 1, with the computational process for each $l$-th pooling layer comprising three main steps. **First**, for the input graph $G^{(l)}$, we assign the set of nodes $V^{(l)}$ completely into $N_{l+1}$ clusters, resulting in a hard node assignment matrix $S_H^{(l)} \in \mathbb{R}^{N_l \times N_{l+1}}$. **Second**, by using the node-based attention mechanism, we compute weights for the nodes within each cluster and aggregate them to form the compressed node feature matrix $X^{(l+1)} \in \mathbb{R}^{N_{l+1} \times d_l}$ for the coarsened graph $G^{(l+1)}$. **Third**, we apply the edge-based attention mechanism to assign weights to each edge, producing the compressed adjacency matrix $A^{(l+1)} \in \mathbb{R}^{N_{l+1} \times N_{l+1}}$ for the next layer. Details of each step are defined as follows.

**Definition 1 (The Hard Node Assignment).** To assign each node exclusively to a single cluster, we first employ the MD-MPNN model (as shown in Eq.(16)) to compute the assignment matrix $S^{(l)} \in \mathbb{R}^{N_l \times N_{l+1}}$ as

$$S^{(l)} = \text{MD-MPNN}_{l,\text{pool}}(A^{(l)}, X^{(l)}). \qquad (1)$$

where $S_{ij}^{(l)}$ represents the probability that the $i$-th node is assigned to the $j$-th cluster. Subsequently, we transform this matrix by setting the maximum value in each row of $S^{(l)}$ to 1 and all other values to 0, resulting in the hard assignment matrix $S_H^{(l)} \in \{0, 1\}^{N_l \times N_{l+1}}$, which ensures that each node is uniquely assigned to the cluster with the highest probability. The $(i, j)$-th entry $S_H^{(l)}(i, j)$ satisfies

$$loc_i^{(l)} = \text{argmax}_j(S_i^{(l)}), \qquad (2)$$

and

$$S_H^{(l)}(i, j) = \begin{cases} 1 & \text{if } j = loc_i^{(l)}; \\ 0 & \text{otherwise.} \end{cases} \qquad (3)$$

It is important to note that for these non-differentiable operations, we apply the straight-through estimator (STE) algorithm (Yin et al., 2019) to enable gradient backpropagation.

**Definition 2 (The Node-based Attention Mechanism).** The process of this computational step is illustrated in Figure 3. Specifically, we first calculate the importance of each node in the entire graph $G^{(l)}$. The input consists of a set of node embeddings $Z^{(l)} = \{\vec{z}_1^{(l)}, \vec{z}_2^{(l)}, ..., \vec{z}_{N_l}^{(l)}\}$, which are also computed by a specified MD-MPNN architecture in Eq.(16) as

$$Z^{(l)} = \text{MD-MPNN}_{l,\text{embed}}(A^{(l)}, X^{(l)}), \qquad (4)$$

where $\vec{z}_i^{(l)} \in \mathbb{R}^{d_l}$. To obtain sufficient expressive power, we commence by transforming the input embeddings $Z^{(l)}$ into more meaningful representations. This is achieved by

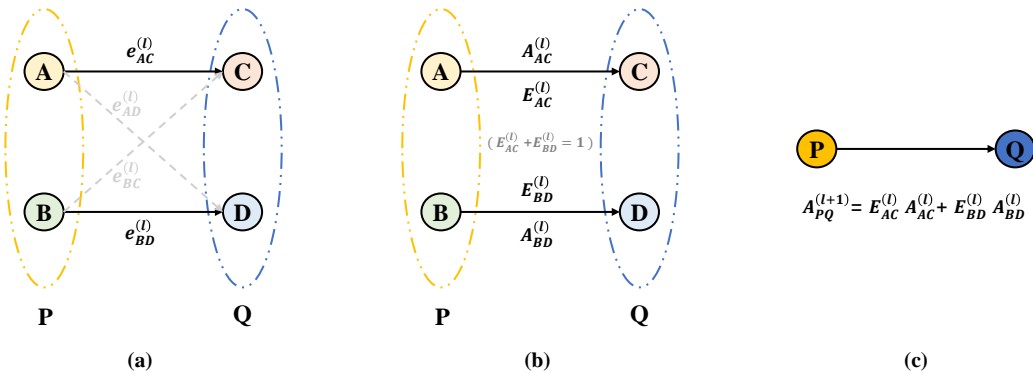

*Figure 4.* The aggregation of edge connectivity strengths via edge-based attention mechanism.

applying a shared linear transformation to every node, parameterized by the weight matrix $\mathbf{W}_m^{(l)} \in \mathbb{R}^{d_l' \times d_l}$, where $d_l'$ denotes the dimensionality of the transformed embeddings. To introduce nonlinearity, the high-level embeddings are then passed through a LeakyReLU activation function with a negative input slope of $\alpha = 0.2$. Subsequently, a self-attention mechanism is employed, which can be implemented using a fully connected layer parameterized by $\vec{\mathbf{a}}_m^{(l)} \in \mathbb{R}^{d_l'}$. Accordingly, the importance of the $i$-th node in the entire graph $m_i^{(l)}$ can be calculated as

$$m_i^{(l)} = \vec{\mathbf{a}}_m^{(l)T} \text{LeakyReLU}(\mathbf{W}_m^{(l)} \vec{z}_i^{(l)}). \qquad (5)$$

Then, we normalize the importance of nodes assigned to the same cluster by applying the softmax function as

$$\alpha_i^{(l)} = \text{softmax}_p(m_i^{(l)}). \qquad (6)$$

where $p$ denotes the $p$-th cluster, and $i$ represents the $i$-th node assigned to that cluster. Finally, we obtain the $p$-th cluster representation $\vec{z}_p^{(l+1)}$ of the coarsened graph $G^{(l+1)}$ by computing the weighted sum of the node embeddings within the $p$-th cluster, using their attention coefficients $\alpha_i^{(l)}$

$$\vec{z}_p^{(l+1)} = \sum_{i \in p} \alpha_i^{(l)} \vec{z}_i^{(l)}. \qquad (7)$$

The resulting cluster representation matrix, i.e., node feature matrix $X^{(l+1)} \in \mathbb{R}^{N_{l+1} \times d_l}$ of the coarsened graph $G^{(l+1)}$ can be computed by row-wisely concatenating all the coarsened node features as

$$X^{(l+1)} = ||_{p=1}^{N_{l+1}} \vec{z}_p^{(l+1)} = ||_{p=1}^{N_{l+1}} \sum_{i \in p} \alpha_i^{(l)} \vec{z}_i^{(l)}. \qquad (8)$$

**Definition 3 (The Edge-based Attention Mechanism).** The process of this computational step is shown in Figure 4. Specifically, we first utilize a weight matrix $\mathbf{W}_e^{(l)} \in \mathbb{R}^{d_l' \times d_l}$ and a self-attention mechanism $a_e^{(l)}$ to calculate the attention

between each pair of nodes $e^{(l)} \in \mathbb{R}^{N_l \times N_l}$ as

$$e_{ij}^{(l)} = a_e^{(l)}(\mathbf{W}_e^{(l)} \vec{z}_i^{(l)}, \mathbf{W}_e^{(l)} \vec{z}_j^{(l)}). \qquad (9)$$

Subsequently, we treat these values as edge attention coefficients and normalize the coefficients of actual edges belonging to the same pair of clusters as

$$E_{ij}^{(l)} = \text{softmax}_{pq}(e_{ij}^{(l)}) = \frac{\exp(e_{ij}^{(l)})}{\sum_{x \in p, y \in q} \exp(e_{xy}^{(l)})}, \qquad (10)$$

where the $i$-th and $j$-th nodes belong to the $p$-th and $q$-th clusters, respectively. Since the attention mechanism $a_e^{(l)}$ can be implemented as a fully connected layer, parameterized by a weight vector $\vec{\mathbf{a}}_e^{(l)} \in \mathbb{R}^{2d_l'}$. The resulting edge attention coefficients $E_{ij}^{(l)}$ can be defined as

$$E_{ij}^{(l)} = \frac{\exp\left(\text{LeakyReLU}\left(\vec{\mathbf{a}}_e^{(l)T}[\mathbf{W}_e^{(l)} \vec{z}_i^{(l)} \| \mathbf{W}_e^{(l)} \vec{z}_j^{(l)}]\right)\right)}{\sum_{x \in p, y \in q} \exp\left(\text{LeakyReLU}\left(\vec{\mathbf{a}}_e^{(l)T}[\mathbf{W}_e^{(l)} \vec{z}_x^{(l)} \| \mathbf{W}_e^{(l)} \vec{z}_y^{(l)}]\right)\right)}. \qquad (11)$$

Finally, we utilize these edge attention coefficients as weights to aggregate the edge connectivity strengths between each pair of clusters, resulting in the adjacency matrix $A^{(l+1)}$ of the coarsened graph $G^{(l+1)}$ as

$$A^{(l+1)} = S_H^{(l)T}(E^{(l)} \odot A^{(l)}) S_H^{(l)}. \qquad (12)$$

Moreover, it is crucial to mention that we intentionally avoid using the edge-based attention mechanism in the first pooling layer. The reason for this is that applying it would result in the connectivity strengths of all edges being consistently 1 after each pooling step, which will lead to the loss of valuable information contained in the edges.

### 3.3. The MD-MPNN Architecture

Graph pooling operations function in conjunction with MPNNs, as they rely on the node embeddings generated

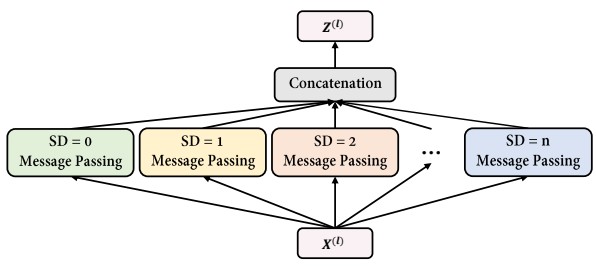

Figure 5. The architecture of MD-MPNN, where **SD=n Message-Passing** indicates that message passing occurs from all nodes that are at a **Shortest Distance** of $n$ from the target node.

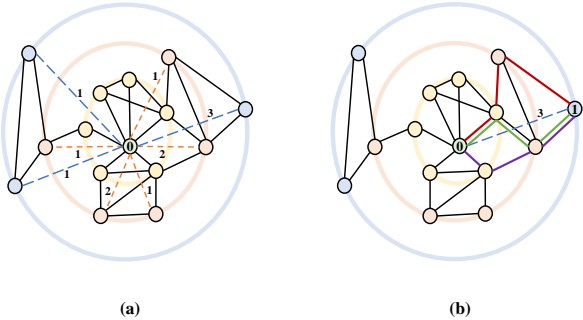

Figure 6. (a) An illustration of the graph rewiring process. Each color representing a different shortest distance from node 0. For SD=n Message-Passing, direct connections are established between the target node and all nodes whose shortest distance to it is $n$, whose connectivity strength is the number of shortest paths. (b) The strength of a direct connection is determined by the number of shortest paths. In this example, the shortest distance between nodes 0 and 1 is 3, because there are three different shortest paths from node 0 to node 1, highlighted in red, green, and purple.

by MPNNs. To address the over-squashing issue in conventional MPNNs and to effectively leverage the edge connectivity strength information in the coarsened graphs produced by ENAHPool, we propose a novel MD-MPNN architecture. The architecture of MD-MPNN is illustrated in Figure 5, and the graph rewiring process is shown in Figure 6.

Specifically, $A^{(l)^h}$ is obtained by raising $A^{(l)}$ to the power of $h$, where $A^{(l)^h}(i,j)$ represents the number of $h$-step random walk paths from the $i$-th node to the $j$-th node in the graph. In order to propagate message on nodes at different shortest distances, we introduce a mask mechanism to filter the connections in $A^{(l)^h}$. To this end, we commence by computing a family of binary matrices $R_h^{(l)} \in \{0,1\}^{N_l \times N_l}$, where the $(i,j)$-th entry $R_h^{(l)}(i,j)$ satisfies

$$R_h^{(l)}(i,j) = \begin{cases} 1 & \text{if } A^{(l)^h}(i,j) > 0; \\ 0 & \text{otherwise.} \end{cases} \tag{13}$$

where $h$ varies from 0 to $H$, and $H$ is a hyperparameter that represents the number of MD-MPNN layers. $R_h(i,j) = 1$

indicates that the random walk departing from the $i$-th node can arrive at the $j$-th node within $h$ steps, i.e., may also include less than $h$ steps. Moreover, we compute another family of binary matrices $U_h^{(l)} \in \{0,1\}^{N_l \times N_l}$ as

$$U_h^{(l)} = \text{CLIP}[R_h^{(l)} - \Sigma_{x=0}^{h-1} R_x^{(l)}], \tag{14}$$

where the function $\text{CLIP}[*]$ is used to restrict the values in $U_h$ to the range $[0,1]$. Specifically, $U_h(i,j) = 1$ indicates that there exists a random walk from the $i$-th node to the $j$-th node with a walk length of at least $h$. In other words, it means that there are paths with a shortest distance of $h$ between the $i$-th and $j$-th nodes. With $U_h^{(l)}$ available, we construct the new topology structures $T_h^{(l)}$ for $G^{(l)}$ using $U_h^{(l)}$ as masks for $A^{(l)^h}$, i.e.,

$$T_h^{(l)} = U_h^{(l)} \odot A^{(l)^h}. \tag{15}$$

As a result, $T_h$ contains only the edges between node pairs with a shortest path distance of $h$, where the edge connectivity strength reflects the number of such paths between them. A greater number of paths suggests a stronger influence on information propagation between nodes. With $T_h^{(l)}$ ($0 \leq h \leq H$) available, the MD-MPNN architecture can be defined in terms of the node feature matrix $X^{(l)}$ and the adjacency matrix $A^{(l)}$ as

$$\text{MD-MPNN}(A^{(l)}, X^{(l)}) = ||_{h=0}^{H} \text{MPNN}(T_h^{(l)}, X^{(l)}). \tag{16}$$

where the backbone of MPNN must aggregate neighborhood information directly from the adjacency matrix (e.g., GCN and GIN). Other variants, such as GAT, tend to ignore the edge connectivity strength information in the graph structure, potentially leading to suboptimal performance.

### 3.4. The Computational Complexity

In our proposed method, the complexity of the MD-MPNN is $O(N^3)$, primarily due to adjacency matrix multiplications. The complexities of the node and edge attention mechanisms are both $O(N)$, as they are computed based on node embeddings. The pooling operation incurs a time complexity of $O(KN^2)$, where $K$ is the number of nodes in the next pooling layer, typically set to $rN$, with $r$ denoting the pooling ratio. Overall, the method maintains a computational complexity of $O(N^3)$, which is comparable to that of existing cluster-based hierarchical pooling approaches.

## 4. Experiments

We empirically compare the proposed method with other deep learning approaches for graph classification across eight benchmark datasets: D&D (Dobson & Doig, 2003), PROTEINS (Borgwardt et al., 2005), NCI1 (Wale et al.,

*Table 1.* Dataset Statistics.

| Dataset | Graphs | Classes | Vertices(Avg.) | Edges(Avg.) | Diameter(Avg.) | Clustering coefficient(Avg) | Labels/Attributes | Domain |
|---|---|---|---|---|---|---|---|---|
| **D&D** | 1178 | 2 | 284.32 | 715.66 | 18.73 | 0.480 | Labels | Biochemical |
| **PROTEINS** | 1113 | 2 | 39.06 | 72.82 | 11.59 | 0.513 | Labels | Biochemical |
| **NCI1** | 4110 | 2 | 29.87 | 32.30 | 13.43 | 0.003 | Labels | Biochemical |
| **FRANKENSTEIN** | 4337 | 2 | 16.90 | 17.88 | 8.45 | 0.011 | Attributes | Biochemical |
| **IMDB-B** | 1000 | 2 | 19.77 | 96.53 | 1.86 | 0.947 | - | Social |
| **IMDB-M** | 1500 | 3 | 13.00 | 65.94 | 1.47 | 0.969 | - | Social |
| **COLLAB** | 5000 | 3 | 74.49 | 2457.78 | 1.86 | 0.891 | - | Social |
| **REDDIT-B** | 2000 | 2 | 429.63 | 497.75 | 8.49 | 0.059 | - | Social |

*Table 2.* Classification Accuracy (In % $\pm$ Standard Error) for Comparisons.[1]

| | D&D | PROTEINS | NCI1 | FRANKENSTEIN | IMDB-B | IMDB-M | COLLAB | REDDIT-B |
|---|---|---|---|---|---|---|---|---|
| Set2Set | 71.94 ± 0.56 | 73.27 ± 0.85 | 68.55 ± 1.92 | 61.46 ± 0.47 | 72.90 ± 0.75 | 50.19 ± 0.39 | 79.55 ± 0.39 | - |
| SortPool | 75.58 ± 0.72 | 73.17 ± 0.88 | 73.82 ± 1.96 | 63.44 ± 0.65 | 72.12 ± 1.12 | 48.18 ± 0.83 | 77.87 ± 0.47 | 76.02 ± 1.73 |
| SAGPool(G) | 71.54 ± 0.91 | 72.02 ± 1.08 | 74.18 ± 1.20 | 62.57 ± 0.60 | 72.16 ± 0.88 | 49.47 ± 0.56 | 78.85 ± 0.56 | 74.45 ± 1.73 |
| GMT | 78.72 ± 0.59 | 75.09 ± 0.59 | 76.35 ± 2.62 | 62.69 ± 0.25 | 73.48 ± 0.76 | 50.66 ± 0.82 | 80.74 ± 0.54 | - |
| DiffPool | 77.56 ± 0.41 | 73.03 ± 1.00 | 62.32 ± 1.90 | 60.60 ± 1.62 | 73.14 ± 0.70 | 51.31 ± 0.72 | 78.68 ± 0.43 | 82.12 ± 1.06 |
| SAGPool(H) | 74.72 ± 0.82 | 71.56 ± 1.49 | 67.45 ± 1.11 | 61.73 ± 0.76 | 72.55 ± 1.28 | 50.23 ± 0.44 | 78.03 ± 0.31 | 75.53 ± 3.53 |
| TopKPool | 73.63 ± 0.55 | 70.48 ± 1.01 | 67.02 ± 2.25 | 61.46 ± 0.84 | 71.58 ± 0.95 | 48.59 ± 0.72 | 77.58 ± 0.85 | 85.12 ± 0.34 |
| ASAP | 76.58 ± 1.04 | 73.92 ± 0.63 | 71.48 ± 0.42 | 66.26 ± 0.47 | 72.81 ± 0.50 | 50.78 ± 0.75 | 78.64 ± 0.50 | - |
| StructPool | 78.45 ± 0.40 | 75.16 ± 0.86 | 78.64 ± 1.53 | - | 72.06 ± 0.64 | 50.23 ± 0.53 | 77.27 ± 0.51 | - |
| MinCutPool | 78.22 ± 0.54 | 74.72 ± 0.48 | 74.25 ± 0.86 | 61.65 ± 0.72 | 72.65 ± 0.75 | 51.04 ± 0.70 | 80.87 ± 0.34 | - |
| SEP-G | 77.98 ± 0.57 | 76.42 ± 0.39 | 78.35 ± 0.33 | - | 74.12 ± 0.56 | 51.53 ± 0.65 | **81.28 ± 0.15** | - |
| ABDPool | 74.13 ± 0.52 | 73.24 ± 0.91 | 71.54 ± 1.28 | - | 70.58 ± 0.71 | 50.63 ± 1.47 | - | 82.75 ± 0.82 |
| **ENAHPool(-MD)**[2] | 79.21 ± 0.51 | 77.06 ± 0.09 | 78.93 ± 0.47 | 65.43 ± 0.21 | 74.44 ± 0.23 | 51.26 ± 0.38 | 78.37 ± 0.24 | 85.60 ± 1.06 |
| **ENAHPool** | **79.91 ± 0.25** | **77.12 ± 0.15** | **79.34 ± 0.31** | **67.68 ± 0.13** | **74.54 ± 0.42** | **51.74 ± 0.16** | 81.09 ± 0.51 | **88.56 ± 0.25** |

[1] The best and second-best results on each dataset in bold and underlined respectively.

[2] **ENAHPool(-MD)** indicates the use of conventional MPNNs for computing node embeddings and assignment matrices.

*Table 3.* The Grid Search Space for the Hyperparameters.

| Hyperparameter | Range |
|---|---|
| Pooling ratio | 0.125, 0.25, 0.5 |
| Pooling layer | 1, 2, 3 |
| MPNN layer | 3, 4, 5, 6, 7 |

2008), FRANKENSTEIN (Orsini et al., 2015), IMDB-B, IMDB-M, COLLAB, and REDDIT-B (Yanardag & Vishwanathan, 2015). Detailed statistics for these datasets are provided in Table 1. Note that when nodes in a graph lack labels or attributes, the node degree can be used as the label.

### 4.1. Baselines and Experimental Settings

We adopt eight hierarchical pooling methods as baselines for comparison, including DiffPool (Ying et al., 2018), SAGPool(H) (Lee et al., 2019), TopKPool (Gao & Ji, 2019), ASAP (Ranjan et al., 2020), StructPool (Yuan & Ji, 2020), MinCutPool (Bianchi et al., 2020), SEP-G (Wu et al., 2022), and ABDPool (Liu et al., 2022). Moreover, we consider four global pooling methods for comparison: Set2Set (Vinyals et al., 2016), SortPool (Zhang et al., 2018), SAGPool(G) (Lee et al., 2019), and GMT (Baek et al., 2021).

In our experiments, we employ 10-fold cross-validation for evaluation and report the average accuracy along with the standard deviation over 10 runs. For the proposed model, we perform hyperparameter tuning using a grid search strategy, as detailed in Table 3. Moreover, to ensure a fair comparison with other methods, the backbone of MPNNs is GCN.

### 4.2. Results and Discussions

Table 2 demonstrates that the proposed ENAHPool operation associated with the MD-MPNN architecture outperforms all alternative methods on seven of the eight datasets. Additionally, even when the MD-MPNN architecture is omitted and conventional MPNNs are used to compute node embeddings and assignment matrices, the single ENAHPool operation still performs better than most alternative methods. The effectiveness can be attributed to two key factors.

**First**, ENAHPool is the first cluster-based hierarchical pooling method to introduce a novel attention mechanism that simultaneously captures the importance of both nodes within clusters and edges between clusters. This effectively addresses the limitation of uniform edge-node information aggregation in existing approaches, resulting in more informative hierarchical representations. **Second**, the MD-MPNN architecture not only alleviates the notorious over-squashing problem in conventional MPNNs, but also leverages the edge connectivity strength information computed by the edge attention mechanism, further enhancing the representational power of the model.

Table 4. Classification Accuracy (In % ± Standard Error) for validating the effectiveness of node assignment strategy.

| Strategy | D&D | PROTEINS | NCI1 | FRANK. | IMDB-B | IMDB-M | COLLAB | REDDIT-B |
|---|---|---|---|---|---|---|---|---|
| Soft assignment | $77.33 \pm 0.43$ | $75.49 \pm 0.03$ | $77.08 \pm 0.38$ | $63.80 \pm 0.04$ | $72.73 \pm 0.49$ | $51.13 \pm 0.35$ | $76.67 \pm 0.14$ | $82.12 \pm 1.06$ |
| Hard assignment | $\mathbf{78.34 \pm 0.62}$ | $\mathbf{76.54 \pm 0.13}$ | $\mathbf{78.22 \pm 0.91}$ | $\mathbf{65.24 \pm 0.18}$ | $\mathbf{73.92 \pm 0.05}$ | $\mathbf{51.18 \pm 0.07}$ | $\mathbf{78.06 \pm 0.25}$ | $\mathbf{83.31 \pm 1.01}$ |

Table 5. Classification Accuracy (In % ± Standard Error) for validating the effectiveness of attention mechanisms.

| Node Att. | Edge Att. | D&D | PROTEINS | NCI1 | FRANK. | IMDB-B | IMDB-M | COLLAB | REDDIT-B |
|---|---|---|---|---|---|---|---|---|---|
| $\times$ | $\times$ | $79.20 \pm 0.48$ | $76.63 \pm 0.21$ | $78.68 \pm 0.65$ | $65.32 \pm 0.62$ | $74.21 \pm 0.64$ | $51.27 \pm 0.09$ | $78.29 \pm 0.49$ | $88.15 \pm 0.15$ |
| $\checkmark$ | $\times$ | $79.50 \pm 0.52$ | $76.73 \pm 0.33$ | $78.85 \pm 0.42$ | $66.09 \pm 0.58$ | $74.31 \pm 1.80$ | $51.60 \pm 1.00$ | $79.38 \pm 0.13$ | $88.21 \pm 0.51$ |
| $\times$ | $\checkmark$ | $79.88 \pm 0.16$ | $76.79 \pm 0.09$ | $79.22 \pm 0.51$ | $66.03 \pm 0.16$ | $74.29 \pm 0.08$ | $51.43 \pm 0.09$ | $78.80 \pm 0.17$ | $88.38 \pm 0.41$ |
| $\checkmark$ | $\checkmark$ | $\mathbf{79.91 \pm 0.25}$ | $\mathbf{77.12 \pm 0.15}$ | $\mathbf{79.34 \pm 0.31}$ | $\mathbf{67.38 \pm 0.13}$ | $\mathbf{74.54 \pm 0.42}$ | $\mathbf{51.74 \pm 0.16}$ | $\mathbf{81.09 \pm 0.51}$ | $\mathbf{88.56 \pm 0.25}$ |

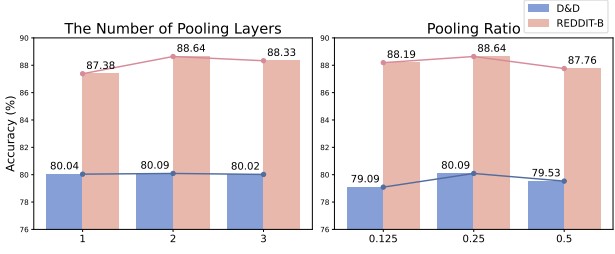

Figure 7. Sensitivity analysis of the number of pooling layers and the pooling ratio.

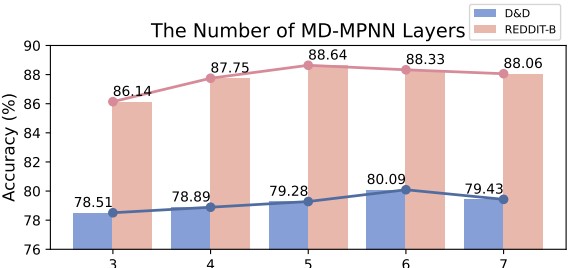

Figure 8. Sensitivity Analysis of the number of MD-MPNN layers.

### 4.3. The Ablation Study

To further evaluate the effectiveness of each module in the proposed ENAHPool operation, we conduct ablation studies on all datasets and report the average accuracy and standard deviation over 5 runs of 10-fold cross-validation.

Specifically, Table 4 presents the impact of the hard node assignment strategy (note that neither setting employs attention mechanisms), and the results demonstrate its effectiveness. Table 5 evaluates the contributions of both the node and edge attention mechanisms. The results indicate that each mechanism individually improves performance, with the best results achieved when they are combined.

### 4.4. Hyperparameter Sensitivity Analysis

In this section, we perform three sensitivity analyses to evaluate the impact of hyperparameters on the proposed ENAHPool operation associated with the MD-MPNN architecture, using two representative datasets: D&D (from the biochemical domain with node labels) and REDDIT-B (from the social domain without node labels or attributes).

First, we examine the effect of the number of pooling layers ($L \in \{1, 2, 3\}$) and the pooling ratios ($k \in \{0.125, 0.25, 0.5\}$). The results in Figure 7 indicate that model performance slightly deteriorates when more than two pooling layers are used, likely due to overfitting. Fur-

thermore, both excessively large and small pooling ratios negatively affect performance. Over-compression leads to a loss of structural information, whereas insufficient compression results in redundancy.

Additionally, we examine the effect of the number of MD-MPNN layers ($H \in \{3, 4, 5, 6, 7\}$). The results in Figure 8 suggest that increasing the number of MD-MPNN layers improves model performance, indicating its effectiveness in alleviating the over-squashing issue in MPNNs. However, beyond a certain threshold, performance begins to decline slightly, which could be attributed to overfitting. Moreover, the optimal number of MD-MPNN layers depends on the graph size, larger graphs require more layers to capture long-range node information.

## 5. Conclusion

In this paper, we have proposed a novel cluster-based hierarchical pooling operation (i.e., ENAHPool) for graph classification. Our method employs attention mechanisms to aggregate node features within clusters and edge connectivity strengths between clusters, resulting in more informative hierarchical representations. Moreover, we develop a new MD-MPNN architecture to effectively alleviate the over-squashing problem arising in MPNNs, further enhancing the model performance. Experimental results indicate the effectiveness of the proposed method.

## Acknowledgement

This work is supported by the National Natural Science Foundation of China under Grants T2122020, 61602535, and 62172370. This work is also supported in part by the Jinhua Science and Technology Plan (No. 2023-3-003a), and the Program for Innovation Research in the Central University of Finance and Economics.

## Impact Statement

This paper presents work whose goal is to advance the field of Machine Learning. There are many potential societal consequences of our work, none which we feel must be specifically highlighted here.

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
