# OpenReview forum: "ENAHPool: The Edge-Node Attention-based Hierarchical Pooling for Graph Neural Networks"
_ICML.cc/2025/Conference — ICML 2025 poster_

### Official Review · Reviewer_FuAf · 2025-02-19

**Overall Recommendation:** 1

**Summary:**

The paper proposes a methodology to perform hierarchical pooling in GNNs along with a message-passing layer that aims at reducing oversquashing (actually oversmoothing).

**Claims And Evidence:**

No. There is an ablation study but I don't feel it covers all the claims and components of the proposed methodology.
Anyway, that is not the main issue.

**Essential References Not Discussed:**

There are several references missing.

**Experimental Designs Or Analyses:**

The experimental evaliuation is rather standard.

**Methods And Evaluation Criteria:**

Yes

**Other Comments Or Suggestions:**

N/A

**Other Strengths And Weaknesses:**

All the methods proposed in this paper are not new and appear in one or more papers from the GNN literature of the last 9 years.
These include:
- using straight-through estimator to make S binary
- using attention in pooling (see e.g., Understanding attention and generalization in graph neural networks from 2019)
- Using attention mechanism to weight the edges (see GAT)
- using heat kernels and random walk in message passing (e.g., Diffusion Improves Graph Learning from 2019)

As is, the paper is more a collection of tricks and optimizations rather than a proposal of a new idea and methodology. That would be OK if this was an applied work to solve a specific problem, which is not the case.
I do not see a contribution to the basic ML research.

**Questions For Authors:**

N/A

**Relation To Broader Scientific Literature:**

The components of the proposed methodology are not novel as they appear in one or more papers from the GNN literature of the last 9 years.

**Theoretical Claims:**

There are no theoretical contributions in this work.

---

> ### Author Rebuttal · Authors · 2025-03-31
>
> Q1: There is an ablation study but I don't feel it covers all the claims and components of the proposed methodology.
>
> A1: Due to space limitations, please refer to our response Q2 to Reviewer foip.
>
> Q2: The contribution to the basic ML research.
>
> A2: This paper proposes a new graph pooling that can overcomes the drawbacks of many existing pooling methods. Specifically, we would like to clarify that GNNs are always a crucial research direction for machine learning, and their development relies on continuously optimizing existing technologies.
>
> Specifically, graph pooling is an essential subfield of GNNs, and most existing methods still suffer from some theoretically drawbacks. For instance, global pooling methods fail to capture hierarchical structural features of graphs. Hierarchical pooling methods based on Top-K selection discard a significant amount of node information, influencing the connectivity construction of the coarsened graph. Cluster-based hierarchical pooling methods can solve the above problems, but they typically aggregate node features and edge connectivity strengths using simple summation, ignoring the differences between nodes and edges. Thus, we propose applying attention mechanisms for weighted aggregation, resulting in more effective coarsened graph representations.
>
> The attention mechanism has been widely applied across various fields. Its core idea is to dynamically assign weights to capture key information, enhancing the model's representation capability. Recently, attention mechanisms have gained traction in graph tasks. For instance, [1] and [2] employ multi-head attention to assess neighbor importance, enabling differentiated aggregation for effective node representations. In graph pooling, [3] uses soft attention to weight node importance, while [4] applies multi-head attention to identify task-relevant parts of the input data and learn each node’s global significance after pooling. [5] integrates attention into second-order pooling, further boosting model expressiveness. Inspired by these works, we introduce a node attention mechanism akin to the self-attention mechanism  in [6], whose success highlights its ability to focus on graph-relevant nodes, providing a solid theoretical foundation for our approach.
>
> Among existing methods, edge attention mechanisms remain underexplored. A notable example is [7], which leverages attention to compute edge contraction scores. Another approach, [8], suggests assigning higher importance to edges connecting more different nodes. Inspired by these works, we introduce an edge attention mechanism akin to the edge information scoring function in [9]. The success of [9] demonstrates its ability to identify important edges in a graph, providing a strong theoretical foundation for our edge attention aggregation strategy.
>
> However, it is important to note that while GAT can be seen as an exploration of edge attention mechanisms, it essentially computes node attention to selectively aggregate neighboring node information, without directly processing edges. This differs from ours, which is designed to aggregate edge connectivity strengths between clusters. The connectivity strengths in the coarsened graph reflects the influence of the information propagation between clusters, determined by both edge importance and quantity. For instance, in a social network, the number of edges between the leadership teams of Companies A and B may be the same as that between the employees of Companies A and C, yet B and C’s influence on A could differ. Our edge attention mechanism aims to capture such differences.
>
> In conclusion, we believe this study provides both practical improvements and a new perspective for theoretical research in machine learning.
>
> [1] Velickovic, P. et al. Graph attention networks. In *ICLR*, 2018. URL https://openreview.net/forum?id=rJXMpikCZ.
>
> [2] Zhang, J. et al. Gaan: Gated attention networks for learning on large and spatiotemporal graphs. In *UAI*, pp. 339–349, 2018.
>
> [3] Li, Y. et al. Gated graph sequence neural networks. In *ICLR*, 2016. URL http://arxiv.org/abs/1511.05493.
>
> [4] Xu Y. et al. Multistructure graph classification method with attention-based pooling. In *IEEE TCSS*, 10(2): 602-613, 2022.
>
> [5] Wang Z. et al. Second-Order Pooling for Graph Neural Networks. In *TPAMI*, vol. 45, no. 6, pp. 6870-6880, 2023.
>
> [6] Lee, J. et al. Self-attention graph pooling. In *ICML*, volume 97 of *Proceedings of Machine Learning Research*, pp. 3734–3743, 2019.
>
> [7] Diehl, F. et al. Towards graph pooling by edge contraction. In *ICML workshop*, 2019.
>
> [8] Gao, Z. et al. Lookhops: light multi-order convolution and pooling for graph classification. *CoRR*, 2020. URL https://arxiv.org/abs/2012.15741.
>
> [9] Yu H. et al. Not all edges are peers: Accurate structure-aware graph pooling networks. In *NN*, 156: 58-66, 2022.
>
> Q3: There are several references missing.
>
> A3: Due to space limitations, please refer to our response Q1 to Reviewer PjzN.

---

> > ### Comment · Reviewer_FuAf · 2025-04-04
> >
> > Thanks for the detailed answer to my review.
> >
> > However, I still believe this is an engineering contribution which is more suitable for, e.g., a Kaggle competition or an applied venue such as KDD rather than a conference like ICML.
> >
> > If the edge attention mechanism is the main focus here, the authors should have introduced only that new component and spent time demonstrating in detail its advantages compared to existing techniques, by providing some theoretical results and controlled experiments to study the different properties of such an operation.
> >
> > But that's not the case. The paper, instead, proposes a methodology with too many other components such as
> >
> > - the straight-through estimator,
> > - the attention on the nodes,
> > - the message passing based on heat kernels.
> >
> > All these components distract and take space from what should be the main focus of the paper (i.e., the edge-attention mechanism for graph pooling) and make it difficult to isolate the single contributions of each individual part.
> >
> > The takeaway message I get from this paper is that edge-attention alone in graph pooling **does not work** and many other tricks and engineering are needed to make the method work.
> >
> > On top of that, edge-oriented pooling methods already exist and are used by the community (https://pytorch-geometric.readthedocs.io/en/latest/generated/torch_geometric.nn.pool.EdgePooling.html), which questions the originality of the proposal.

---

> > > ### Author Response · Authors · 2025-04-04
> > >
> > > Thank you for your thorough review. We note that you have deleted the concern about GAT, since there were some misunderstanding between GAT and our method. However, it seems that there may still be some other misunderstandings for our work, and we would like to explain them again.
> > >
> > > First, our paper is not solely focused on the edge attention mechanism, it is just one component of our approach. One of our goal is to address the oversimplified aggregation of both nodes within clusters and edges between clusters in cluster-based hierarchical pooling methods. To this end, we propose the Edge-Node Attention Mechanism. Second, to better leverage the cluster representations and connectivity strengths between clusters obtained through attention-based aggregation, we also introduce MD-MPNN architecture. These components, when combined, constitute our proposed method.
> > >
> > > Furthermore, in terms of the EdgePool, as we have discussed in our previous response, this approach is theoretically different from ours. Specifically, the EdgePool performs pooling by progressively **removing edges** based on computed edge contraction scores, whereas our edge attention mechanism adaptively **aggregates edge** information for pooling. Since **removing edges** and **aggregating edge** are entirly different operations, we believe this distinction underscores the originality of our method. We sincerely hope the reviewer can significantly identify the above theoretically differences.

---

### Official Review · Reviewer_PjzN · 2025-03-07

**Overall Recommendation:** 5

**Summary:**

This paper introduces a novel graph pooling (ENAHPool), by combining the hard node assignment and the attention machnism in an interesting way. Different from other pooling operations, the new ENAHPool can compress the nodes and edges into hierarchical graphs associated with the node and edge attention rather than simply summing them up. The experiments on some standard datasets have demonstrated that the new ENAHPool can significantly improve the graph classification performance for the GNN models.

**Claims And Evidence:**

I think the claims are supported by clear and convincing evidence.

**Essential References Not Discussed:**

The references should be completed, and well support for the new pooling operation.

**Experimental Designs Or Analyses:**

I have checked the experimental setups.

**Methods And Evaluation Criteria:**

The proposed ENAHPool makes sense.

**Other Comments Or Suggestions:**

See my rebuttal questions below.

**Other Strengths And Weaknesses:**

This paper has some strengths, such as
1. This paper is well organized and easy to follow.
2. Different from the current graph pooling operation, the new proposed ENAHPool propose a new edge-based attention to discriminate the importance of the edges between two clusters.
3. To further improve the performance of the proposed ENAHPool, this paper also proposes a MPNN module to directly propagate the node information based on different distances, so that the new pooling operation can avoid the over-squashing problem.
However, I still have some concerns for this paper, please refer to the following rebuttal questions.

**Questions For Authors:**

1.  Can you provide some detailed time complexity analysis for the proposed ENAHPool operation?
2. Excluding the GIN and GCN architecture, I wonder whether the proposed ENAHPool operation can be used for other GNN architecture, and improve the performance?
3. The authors stated that the MD-MPNN medule can address the over-squarshing problem, how about perform the medule for mutiple times?

**Relation To Broader Scientific Literature:**

The new ENAHPool can capture either node and edge attentions to re-weight the importance of the nodes belonging to the same cluster or the edges between tow clusters. So, the new pooling operation can adaptively discriminant the most significant node and edge information. This can provide useful structure feature information for the GNNs.

**Theoretical Claims:**

I have checked.

---

> ### Author Rebuttal · Authors · 2025-03-31
>
> Q1: References for the new pooling operations.
>
> A1: Thank you for your valuable suggestion. We have further investigated more recently proposed pooling operations and will refine the related work section in the final version.
>
> Recent research on graph pooling has primarily focused on cluster-based hierarchical methods. MathNet [1] leverages Haar-like wavelet multiresolution analysis to construct hierarchical graph representations. Tsitsulin et al. [2] introduced Deep Modularity Networks (DMoN), an unsupervised approach that optimizes the clustering process using the modularity measure. Bacciu et al. [3] proposed a pooling method based on Maximal k-Independent Sets (k-MIS), ensuring that selected nodes maintain a minimum pairwise distance of *k*. Zhou et al. [4] proposed Cross-View Graph Pooling, which integrates pooled graph information from both node and edge perspectives. WGDPool [5] utilizes a differentiable *k*-means clustering mechanism with Softmin assignments based on node-centroid distances.
>
> [1] Zheng, X. et al. Mathnet: Haar-like wavelet multiresolution analysis for graph representation learning. *Knowl. Based Syst.*, 273: 110609, 2023.
>
> [2] Tsitsulin, A.et al. Graph clustering with graph neural networks. *J. Mach. Learn. Res.*, 24:127:1–127:21, 2023.
>
> [3] Bacciu, D. et al. Generalizing downsampling from regular data to graphs. In *AAAI*, pp. 6718–6727, 2023.
>
> [4] Zhou, X. et al. Edge but not least: Cross-view graph pooling. In *ECML PKDD*, volume 13714 of *Lecture Notes in Computer Science*, pp. 344–359, 2022.
>
> [5] Xiao, Z. et al. Wgdpool: A broad scope extraction for weighted graph data. *Expert Syst. Appl.*, 249:123678, 2024.
>
> Q2: Detailed time complexity analysis.
>
> A2: In our proposed method, each pooling layer primarily involves the following steps. First, the MD-MPNN model performs convolution operations to obtain node embeddings. The computational complexity of MD-MPNN is $O(N^3)$ due to the need for matrix multiplications on the adjacency matrix to filter node information at different distances, helping to mitigate over-squashing issue. Second, the computational complexity of both node and edge attention mechanism are $O(N)$, as they are computed based on node features. Finally, the pooling operation has a time complexity of $O(KN^2)$, where $K$ represents the number of nodes in the next pooling layer, generally set as $rN$, with $r$ denoting the pooling ratio. Overall, the proposed method maintains an overall computational complexity of $O(N^3)$, which is comparable to other classic cluster-based hierarchical pooling methods such as StructPool [1] and MinCutPool [2].
>
> [1] Yuan, H. et al. Structpool: Structured graph pooling via conditional random fields. In *ICLR*, 2020. URL https://openreview.net/forum?id=BJxg_hVtwH.
>
> [2] Bianchi, F. M. et al. Spectral clustering with graph neural networks for graph pooling. In *ICML*, volume 119 of *Proceedings of Machine Learning Research*, pp. 874–883, 2020.
>
> Q3: Improve the performance of other GNN associated with ENAHPool?
>
> A3: Of course! Our proposed ENAHPool operation is compatible with any GNN architecture, as it focuses on aggregating nodes and edges during pooling, without constraining the choice of GNN. However, it is preferable to use a GNN that aggregates neighborhood information based on the adjacency matrix. Otherwise, it may fail to leverage the edge connectivity strengths between clusters learned by the edge attention mechanism (such as GAT), potentially causing our method to degrade into a purely node attention-based hierarchical pooling operation.
>
> Q4: Address the over-squashing problem when perform the module for multiple times?
>
> A4: To address the over-squashing issue, we analyzed the impact of the number of MD-MPNN layers on model performance (as shown in Figure 7 of the paper). The results demonstrate that, unlike traditional MPNNs, increasing the number of MD-MPNN layers gradually improves model performance, mainly due to its novel message-passing mechanism. However, once the number of layers exceeds a certain threshold, performance begins to decline slightly. This is likely because information from excessively distant nodes introduces noise, which diminishes the effectiveness of feature representation.

---

> > ### Comment · Reviewer_PjzN · 2025-04-05
> >
> > The authors addressed my concerns, and the additional experiments also make the statements more convincing, such as 1) the edge attentions, 2) the combination of edge and node attentions, and 3) the hard node assignment.  All these indicate that the new proposed graph pooling is novel and effective.
> >
> > Overall, I think the new ENAHPool is an important contribution to the Graph ML community, and I am willing to raise my score by one point. I trust this paper may enlighten some new works in future.

---

### Official Review · Reviewer_foip · 2025-03-10

**Overall Recommendation:** 5

**Summary:**

This paper develops a novel graph pooling method, namely the ENAHPoo, for graph classification associated with GNNs. Different from the previous graph pooling methods, the ENAHPool simultaneously integrtes either node or edge attention for the hierarchical sturcutre learning. In addition, it also design an associated MD-MPNN mode to further mitigate the over-squashing problem through different distance relationship. Experimental evaluations demonstrates the effectiveness

**Claims And Evidence:**

Yes, the claims of this paper are supported by clear and convincing evidence.

**Essential References Not Discussed:**

It seems that there is no necessary reference missed. The current references cited in this work provide sufficient contents for the proposed ENAHPool method.

**Experimental Designs Or Analyses:**

Yes, the experimental setting is completer and results/analyses are convincing.

**Methods And Evaluation Criteria:**

Yes, the evaluation makes sense. Specifically, the proposed graph pooling method is very fundamental and important in the field of graph neural networks.

**Other Comments Or Suggestions:**

See the weakness.

**Other Strengths And Weaknesses:**

Strengths:

S1. A novel hierarchical pooling method associated with the node and edge attention, for learning hierarchical structures.

S2. Efficient computational procedures for the node and edge attentions.

S3. reduce the over-squashing problem therough the associated MD-MPNN associated with the ENAHPool.

Weakness:

W1. The abstract is a little long, the authors should briefly summarize the contribution, and make it shorter.

W2. Although the experiments demonstrate the effectiveness, for the Ablation Study, the author only evaluate the performance on two of the datasets for classification performance comparisons. More dataset for this study is prefered, and the author can put them in the supplementary material if the space is not enough.

**Questions For Authors:**

This paper mentioned several times that the classical method is not efficient, however how about the the computational efficiency of the propose methods? I didn’t see any detailed discussion about the issue.

**Relation To Broader Scientific Literature:**

A novel framework to simultaneously capture both the node and edge attention for hierarchical structure learning, i.e. for the resulting coarsened graph by compress the nodes belonging to the same cluster.

**Theoretical Claims:**

I have checked. And, all proofs of theoretical claims are correct.

---

> ### Author Rebuttal · Authors · 2025-03-31
>
> Q1: The abstract is a little long.
>
> A1: We will update the abstract in the final version to make it more concise and easier to understand.
>
> Q2: More dataset is preferred for Ablation Study.
>
> A2: Thank you for your constructive suggestion. We have conducted the ablation experiments on all datasets. However, due to time constraints, we only report the average results from 5 runs of 10-fold cross-validation here.
>
> |Assignment strategy|D&D|PROTEINS|NCI1|FRANK.|IMDB-B|IMDB-M|COLLAB|REDDIT-B|
> |-|-|-|-|-|-|-|-|-|
> |Soft assignment|77.33 $\pm$ 0.43|75.49 $\pm$ 0.03|77.08 $\pm$ 0.38|63.80 $\pm$ 0.04|72.73 $\pm$ 0.49|51.13 $\pm$ 0.35|76.67 $\pm$ 0.14|82.12 $\pm$ 1.06|
> |Hard assignment|**78.34 $\pm$ 0.62**|**76.54 $\pm$ 0.13**|**78.22 $\pm$ 0.91**|**65.24 $\pm$ 0.18**|**73.92 $\pm$ 0.05**|**51.18 $\pm$ 0.07**|**78.06 $\pm$ 0.25**|**83.31 $\pm$ 1.01**|
>
> |Node Att.|Edge Att.|D&D|PROTEINS|NCI1|FRANK.|IMDB-B|IMDB-M|COLLAB|REDDIT-B|
> |-|-|-|-|-|-|-|-|-|-|
> |$\times$|$\times$|79.20 $\pm$ 0.48|76.63 $\pm$ 0.21|78.68 $\pm$ 0.65|65.32 $\pm$ 0.62|74.21 $\pm$ 0.64|51.27 $\pm$ 0.09|78.29 $\pm$ 0.49|88.15 $\pm$ 0.15|
> |$\checkmark$|$\times$|79.50 $\pm$ 0.52|76.73 $\pm$ 0.33|78.85 $\pm$ 0.42|66.09 $\pm$ 0.58|74.31 $\pm$ 1.80|51.60 $\pm$ 1.00|79.38 $\pm$ 0.13|88.21 $\pm$ 0.51|
> |$\times$|$\checkmark$|79.88 $\pm$ 0.16|76.79 $\pm$ 0.09|79.22 $\pm$ 0.51|66.03 $\pm$ 0.16|74.29 $\pm$ 0.08|51.43 $\pm$ 0.09|78.80 $\pm$ 0.17|88.38 $\pm$ 0.41|
> |$\checkmark$|$\checkmark$|**79.91 $\pm$ 0.25**|**77.12 $\pm$ 0.15**|**79.34 $\pm$ 0.31**|**67.38 $\pm$ 0.13**|**74.54 $\pm$ 0.42**|**51.74 $\pm$ 0.16**|**81.09 $\pm$ 0.51**|**88.56 $\pm$ 0.25**|
>
> Q3: Detailed discussion of the computational efficiency.
>
> A3: In our proposed method, each pooling layer primarily involves the following steps. First, the MD-MPNN model performs convolution operations to obtain node embeddings. The computational complexity of MD-MPNN is $O(N^3)$ due to the need for matrix multiplications on the adjacency matrix to filter node information at different distances, helping to mitigate over-squashing issue. Second, the computational complexity of both node and edge attention mechanism are $O(N)$, as they are computed based on node features. Finally, the pooling operation has a time complexity of $O(KN^2)$, where $K$ represents the number of nodes in the next pooling layer, generally set as $rN$, with $r$ denoting the pooling ratio. Overall, the proposed method maintains an overall computational complexity of $O(N^3)$, which is comparable to other classic cluster-based hierarchical pooling methods such as StructPool [1] and MinCutPool [2].
>
> It is worth noting that our paper repeatedly emphasizes the inefficiency of using attention mechanisms for node aggregation in existing cluster-based hierarchical pooling methods. For example, ABDPool [3] computes scaled dot-product self-attention for nodes within each cluster, while C2N-ABDP [4] extracts cluster representations via singular value decomposition and then computes attention between each cluster and its nodes for weighted aggregation. These methods require iterating over every cluster, leading to high computational complexity. In contrast, the attention mechanisms we propose are significantly more computationally efficient.
>
> [1] Yuan, H. et al. Structpool: Structured graph pooling via conditional random fields. In *ICLR*, 2020. URL https://openreview.net/forum?id=BJxg_hVtwH.
>
> [2] Bianchi, F. M. et al. Spectral clustering with graph neural networks for graph pooling. In *ICML*, volume 119 of *Proceedings of Machine Learning Research*, pp. 874–883, 2020.
>
> [3] Liu, Y. et al. Abdpool: Attention-based differentiable pooling. In *ICPR*, pp. 3021–3026,2022.
>
> [4] Ye, R. et al. C2N-ABDP: cluster-to-node attention-based differentiable pooling. In *GbRPR*, volume 14121 of *Lecture Notes in Computer Science*, pp. 70–80, 2023.

---

### Official Review · Reviewer_PecS · 2025-03-13

**Overall Recommendation:** 2

**Summary:**

The paper proposes a cluster-based pooling method for graph neural networks (GNNs). The main feature of the proposal is that it performs an hard assignment of the input nodes, i.e., each node belongs to one cluster. Also, attention mechanism are employed to build node features and adjacency matrix of the coarsened graph.

Additionally, the author proposes a new GNN to mitigate the over-squashing problem of GNNs

The effectiveness of the combination between the proposed pooling and GNN is demonstrated with a set of experiment.

### Update after rebuttal
I would like to keep my score as it is since the author's response does not completely address my main concerns.

I suggest the authors to add more details about the experiments to make sure that the comparison is fair. For example, the proposed architecture employs more parameters (given the same hidden size and number of layers) than the baseline methods. This was also my concern about the hard vs. soft comparison: it is not clear if only the argmax of the assignment matrix $S$  has been removed leaving everything else the same.

Finally, I suggest always including the anonymized code repository in the paper.

**Claims And Evidence:**

- The paper claims that soft-assignment worsens the performance of cluster-based pooling method. Nevertheless, I did not find convincing evidence in the paper's results since Table 4 shows the results on only 2 datasets. I believe that this is a key point that justifies the proposal; thus, the difference between hard and soft assignments should have been computed on all the datasets.
- The paper claims that coarsened graphs are usually built without considering the importance of edges. To this end, it introduces an attention mechanism to build both node features and the structure of the coarsened graph. Nevertheless, the ablation study in Table 5 does not show that employing such a mechanism is always beneficial. For example, by looking at the statistical significance of the differences among the results, it seems that node attention is unnecessary.

In general, I did not understand the idea behind the proposal, and the results were not convincing enough to justify alone the methods proposed.

**Essential References Not Discussed:**

I do not have other papers to suggest.

**Experimental Designs Or Analyses:**

The experimental design is not clear since:
- There is no mention of how the baseline methods have been trained, and how their hyper-parameters selected;
- It is not clear how the data have been split in training validation and test. There is onluy a mention to 10-fold cross-validation , but it is not clear if it is for model assessment or model selection
- There is a mention to auxiliary classifiers during training, but it is not clear how they are used, and if they have been applied also for baseline methods.
- The results of baseline method are different from the ones obtained in other papers. For example, MinCut (Bianchi et al., 2020) reaches an accuracy of 80.8 +/- 2.3 on D&D dataset, that is higher than the one reported in the paper.

**Methods And Evaluation Criteria:**

The evaluation criteria is reasonable, but the proposed methodology is not justified enough.

**Other Comments Or Suggestions:**

Nothing.

**Other Strengths And Weaknesses:**

The paper is clear and well-written.

To the best of my knowledge, the idea is novel. However, the methodology proposed has no substantial basis.

**Questions For Authors:**

Nothing.

**Relation To Broader Scientific Literature:**

I did not carefully check if the key contributions have been discussed adequately with respect to the existing literature.

**Theoretical Claims:**

Ther are no theoretical claims.

---

> ### Author Rebuttal · Authors · 2025-03-31
>
> Q1: Convincing evidence is needed to claims that soft-assignment worsens the performance.
>
> A1: Thanks for the suggestion. We conducted comparative experiments on all datasets to verify the positive impact of the hard assignment operation on model performance. However, due to time constraints, we only report the average results from 5 runs of 10-fold cross-validation here.
>
> |Assignment strategy|D&D|PROTEINS|NCI1|FRANK.|IMDB-B|IMDB-M|COLLAB|REDDIT-B|
> |-|-|-|-|-|-|-|-|-|
> |Soft assignment|77.33 $\pm$ 0.43|75.49 $\pm$ 0.03|77.08 $\pm$ 0.38|63.80 $\pm$ 0.04|72.73 $\pm$ 0.49|51.13 $\pm$ 0.35|76.67 $\pm$ 0.14|82.12 $\pm$ 1.06|
> |Hard assignment|**78.34 $\pm$ 0.62**|**76.54 $\pm$ 0.13**|**78.22 $\pm$ 0.91**|**65.24 $\pm$ 0.18**|**73.92 $\pm$ 0.05**|**51.18 $\pm$ 0.07**|**78.06 $\pm$ 0.25**|**83.31 $\pm$ 1.01**|
>
> Q2: Convincing experiments and discussions are needed to verify the significance of different attention mechanisms.
>
> A2: Theoretically, the aim of using edge-node attentions is to obtain a more meaningful coarsened graph, capturing more accurate hierarchical representations. These two attention mechanisms are indispensable. First, without the node attention, hierarchical pooling would flatten the aggregation of nodes within a cluster, disregarding their varying importance. Second, the edge connectivity strengths between clusters naturally serve as attention during graph convolution. To better leverage this property, we propose an edge attention mechanism, which enhances the aggregation weights of important edges and prevents the neglect of rare but highly valuable connections between each cluster.
>
> The misunderstanding may have arisen because we only presented the ablation results for two datasets in the paper. To clarify this, we conducted comparative experiments on all datasets. However, due to time constraints, we only report the average results from 5 runs of 10-fold cross-validation here. The results show that both the node and edge attention mechanism have a positive impact on model performance, though their degree of influence varies across datasets. Moreover, the best results are consistently achieved when both node and edge attention mechanisms are used together, which demonstrates the effectiveness of our approach.
>
> |Node Att.|Edge Att.|D&D|PROTEINS|NCI1|FRANK.|IMDB-B|IMDB-M|COLLAB|REDDIT-B|
> |-|-|-|-|-|-|-|-|-|-|
> |$\times$|$\times$|79.20 $\pm$ 0.48|76.63 $\pm$ 0.21|78.68 $\pm$ 0.65|65.32 $\pm$ 0.62|74.21 $\pm$ 0.64|51.27 $\pm$ 0.09|78.29 $\pm$ 0.49|88.15 $\pm$ 0.15|
> |$\checkmark$|$\times$|79.50 $\pm$ 0.52|76.73 $\pm$ 0.33|78.85 $\pm$ 0.42|66.09 $\pm$ 0.58|74.31 $\pm$ 1.80|51.60 $\pm$ 1.00|79.38 $\pm$ 0.13|88.21 $\pm$ 0.51|
> |$\times$|$\checkmark$|79.88 $\pm$ 0.16|76.79 $\pm$ 0.09|79.22 $\pm$ 0.51|66.03 $\pm$ 0.16|74.29 $\pm$ 0.08|51.43 $\pm$ 0.09|78.80 $\pm$ 0.17|88.38 $\pm$ 0.41|
> |$\checkmark$|$\checkmark$|**79.91 $\pm$ 0.25**|**77.12 $\pm$ 0.15**|**79.34 $\pm$ 0.31**|**67.38 $\pm$ 0.13**|**74.54 $\pm$ 0.42**|**51.74 $\pm$ 0.16**|**81.09 $\pm$ 0.51**|**88.56 $\pm$ 0.25**|
>
> Q3: Evaluation criteria is reasonable, but the proposed methodology is not justified enough.
>
> Due to space limitations, please refer to our response Q2 to Reviewer FuAf.
>
> Q4: The experimental design is not clear.
>
> To minimize the impact of data partitioning on model evaluation, we followed the testing methodology from [1]. Specifically, we performed 10 random splits of each dataset and conducted 10-fold cross-validation on each split. The model was evaluated on every validation set, resulting in a total of 100 evaluation results for each method on each dataset. The experimental setup of MinCutPool differs from ours. Although it also performs 10 random splits of the dataset, it does not conduct cross-validation. This operation may make the results more sensitive to the partition, which could explain the differences in reported accuracy.
>
> In addition, for the baselines and benchmarks that were tested in [1], we directly cite the reported results, as we follow the same experimental setup. For those not covered in [1], we adhere to the hyperparameter settings specified in the paper for evaluation.
>
> Additionally, regarding the auxiliary classifiers, we sincerely apologize for this oversight. In the initial version of our experiments, we attempted to include this module. However, after further discussion, we decided to remove it to ensure fair comparisons with the baselines. Unfortunately, due to insufficient proofreading of the final manuscript, this change was not properly reflected in the text. We will correct this in the final version and ensure that all experimental descriptions align with our code. If necessary, we can also submit the code with anonymous link, then you can check.
>
> [1] Wu, J. et al. Structural entropy guided graph hierarchical pooling. In *ICML*, volume 162 of *Proceedings of Machine Learning Research*, pp. 24017–24030, 2022.

---

> > ### Comment · Reviewer_PecS · 2025-04-04
> >
> > I thank the authors for their response.
> >
> > However, I still have some concerns:
> >
> > 1. It is not clear what architecture was used for the soft assignments.
> > 2. The table reported here to assess the effectiveness of node/edge attention does not show a clear advantage. We should always consider the standard deviation when we compare results.
> > 3. The paper is almost empirical and there are some tricks in the experiment that can hide the effectiveness of the proposale (e.g. the ENAHPool employes two networks rather than one, the node embeddings of each layer are concatenated)

---

> > > ### Author Response · Authors · 2025-04-04
> > >
> > > Thanks for the feedback. In response to the comments, we would like to clarify the following points.
> > >
> > > Q1: It is not clear what architecture was used for the soft assignments.
> > >
> > > A1: The architecture of soft assignment is formulated as  $S^{(l)}=\text{GNN}(A^{(l)}, X^{(l)})$ that was widely employed for many hierarchical graph pooling operations.
> > >
> > > Q2: The table reported here to assess the effectiveness of node/edge attention does not show a clear advantage. We should always consider the standard deviation when we compare results.
> > >
> > > A2: Even when comparing standard deviations, the method incorporating both node and edge attention still performs the best on many benchmarks, such as NCI1 and FRANK.
> > >
> > > Q3: The paper is almost empirical and there are some tricks in the experiment that can hide the effectiveness of the proposal (e.g. the ENAHPool employes two networks rather than one, the node embeddings of each layer are concatenated)
> > >
> > > A3: It is important to clarify that we **did not use any tricks** in our experiments.
> > >
> > > First, we did not use two separated networks, instead, we simultaneously compute both the node and edge attention at each pooling step. As a result, **each pooling layer** produces only a **single coarsened graph**, where the cluster representations are obtained as the weighted sum of the original node features, and the connectivity strengths between clusters are computed as the weighted sum of the original edge connectivity strengths.
> > >
> > > Second, we are not sure what you mean by "the node embeddings of each layer are concatenated". If you are referring to the message passing layers in MD-MPNN, then yes, we do concatenate the node embeddings from different layers. This is because our goal is to capture neighborhood information at varying distances. In MD-MPNN, the node embedding at layer $i$ can only capture information from neighbors that are exactly $i$ hops away, so concatenating embeddings from different layers allows us to integrate multi-hop neighborhood information effectively.
> > >
> > > If necessary, we can submit the code with an anonymous link, indicating that we did not use any so-called trick for our experiments.

---

### Decision · Program_Chairs · 2025-05-01

**Decision:**

Accept (poster)

**Comment:**

This paper proposes a method called ENAHPool to enhance Graph Neural Networks (GNNs). It performs hard node assignments to clusters and aggregates node features and edge connectivity with learned attention weights, which reduces ambiguity compared to soft assignments.  The dual attention mechanism helps to learn more expressive coarsened graph representations. The paper also proposes a companion architecture called MD-MPNN to alleviate the over-squashing problem. However, there are concerns that it combines too many components, which increases complexity and makes it hard to isolate each contribution. There is no strong theoretical justification or proof provided. Overall, this is a good paper and should be considered if there is room in the program.